# Dibenzocyclooctadiene Lignans in Plant Parts and Fermented Beverages of *Schisandra chinensis*

**DOI:** 10.3390/plants10020361

**Published:** 2021-02-13

**Authors:** Woo Sung Park, Kyung Ah Koo, Ji-Yeong Bae, Hye-Jin Kim, Dong-Min Kang, Ji-Min Kwon, Seung-Mann Paek, Mi Kyeong Lee, Chul Young Kim, Mi-Jeong Ahn

**Affiliations:** 1College of Pharmacy and Research Institute of Pharmaceutical Sciences, Gyeongsang National University, Jinju 52828, Korea; pws8822@gmail.com (W.S.P.); kk408842@gmail.com (K.A.K.); black200203@gmail.com (H.-J.K.); dongminkang71@gmail.com (D.-M.K.); ellusian458@gmail.com (J.-M.K.); million@gnu.ac.kr (S.-M.P.); 2College of Pharmacy, Jeju National University, Jeju 63243, Korea; jybae@jejunu.ac.kr; 3Interdisciplinary Graduate Program in Advanced Convergence Technology & Science, Jeju National University, Jeju 63243, Korea; 4College of Pharmacy, Chungbuk National University, Cheongju 28160, Korea; mklee@chungbuk.ac.kr; 5College of Pharmacy, Institute of Pharmaceutical Science and Technology, Hanyang University, Ansan 15588, Korea

**Keywords:** *Schisandra chinensis*, Omija, dibenzocyclooctadiene lignans, seed, flower, fermented beverage

## Abstract

The fruit of *Schisandra chinensis*, Omija, is a well-known traditional medicine used as an anti-tussive and anti-diarrhea agent, with various biological activities derived from the dibenzocyclooctadiene-type lignans. A high-pressure liquid chromatography-diode array detector (HPLC-DAD) method was used to determine seven lignans (schisandrol A and B, tigloylgomisin H, angeloylgomisin H, schisandrin A, B, and C) in the different plant parts and beverages of the fruit of *S. chinensis* grown in Korea. The contents of these lignans in the plant parts descended in the following order: seeds, flowers, leaves, pulp, and stems. The total lignan content in Omija beverages fermented with white sugar for 12 months increased by 2.6-fold. Omija was fermented for 12 months with white sugar, brown sugar, and oligosaccharide/white sugar (1:1, *w*/*w*). The total lignan content in Omija fermented with oligosaccharide/white sugar was approximately 1.2- and 1.7-fold higher than those fermented with white sugar and brown sugar, respectively. A drink prepared by immersion of the fruit in alcohol had a higher total lignan content than these fermented beverages. This is the first report documenting the quantitative changes in dibenzocyclooctadiene-type lignans over a fermentation period and the effects of the fermentable sugars on this eco-friendly fermentation process.

## 1. Introduction

*Schisandra chinensis* (Turcz.) Baill. (family Schisandraceae) is a deciduous, dioecious, woody vine plant also known as five-flavor-fruit in East Asia [1]. The fruit called Omija in Korea has been used in the treatment of diseases related to the gastrointestinal tract, respiratory failure, cardiovascular diseases, body fatigue, and weakness [2]. Diverse components of this fruit, such as lignans, terpenes, essential oils, anthocyanins, and polysaccharides have demonstrated anticancer, neuroprotective, anti-inflammatory, antioxidative, hepatoprotective, immunomodulatory, and antiviral activities [3]. Among them, dibenzocyclooctadiene-type lignans are well known to show various biological activities that could prevent cancer, hepatotoxicity, and nephrotoxicity in humans [4,5,6]. Schisandrin A–C, gomisin A–H, and schisandrol A and B are the major lignans that are known to exhibit the biological activities described above [6,7,8]. Recently, a study reported on the addition of Omija fruit to ale-type beer at different points in the brewing process in an attempt to develop an Omija fruit beverage with high antioxidant capacity [9].

Thousands of tons of schisandra berries are traded annually in Asia. They are currently commercially consumed as functional health foods in the form of beverages, jams, capsules, and powders. In general, only the fruit is beneficially consumed, and the other parts of the plant are discarded. While previous studies have focused mainly on the content of various chemical components, including lignans, in the fruit of *S. chinensis*, and in the roots or branches in some cases [10,11,12,13,14,15,16], there is no report on the difference in the lignan content of the plant parts, including the flower.

The fermentation of foods has a variety of benefits, including improving the flavor and long-term storage as well as increasing or producing beneficial nutrients and healthy ingredients. The raw berry of *S. chinensis* is converted primarily to a fermented beverage with sugar, as the most common form for consumption in Korea. In general, people make this beverage at home and consume the beverage after 6 or 12 months or a much longer period of fermentation. Fermentation with lactic acid bacteria or with only sugar results in changes in various biological activities, such as the anti-oxidative, anti-inflammatory, anti-hypertensive, and anti-cholesterol activities [17,18,19,20,21]. To the best of our knowledge, this is the first report describing the effects of the fermentation process, fermentation period, and fermenting materials on the lignan content of Omija beverages and drink.

In this study, the content of seven major dibenzocyclooctadiene-type lignans (schisandrol A (schisandrin or wuweizisu A), schisandrol B (gomisin A), tigloylgomisin H, angeloylgomisin H, schisandrin A (deoxyschisandrin), schisandrin B (gomisin N or wuweizisu B or γ-schisandrin), and schisandrin C) in the stem, leaf, pulp, seed, and flower of *S. chinensis* is determined. The content of these compounds in Omija beverages made from the whole fruit and fermented with white sugar for different fermentation times is also determined. The change in the lignan content is compared for fruit beverages fermented with three kinds of sugars (white, brown, and oligosaccharide/white) and ethanol.

## 2. Results and Discussion

### 2.1. Lignan Content in Plant Parts of S. chinensis

Calibration curves were constructed by analyzing a standard mixture containing the seven lignans (Figure 1) at various concentration levels and plotting the peak area against the concentration of each reference standard (Table 1). The curves showed good linearity, and the correlation coefficients were between 0.997 and 0.999 for all the compounds over the concentration ranges of quantification. The recovery of seven lignans was assessed by spiking samples with high and low concentrations (50 and 10 ng, respectively) of each reference compound. The average recoveries were between 84.9% and 105.7% (n = 3). The limits of detection (LOD) were determined by serial dilution based on a signal-to-noise (S/N) ratio of 3:1 (Table 1). Limits of quantification (LOQ) were 1 ng for the seven lignans. The peak purity was determined using a photodiode array detector. In addition, each peak in the absorption spectrum was compared with the characteristic peak of the corresponding standard compound. The precision was determined from the intra-day and inter-day repeatability, and the relative standard deviations were below 19%.

The variation in the content of seven lignans in the various parts of *S. chinensis* was evaluated by HPLC-DAD (Figure 2). The plant parts included the stem, leaf, flower, pulp, and seed. The total lignan content was highest for the seeds, with a value of 47.42 ± 2.81 mg/g·DW (Table 2). This value was more than eight times higher than the lignan contents in the other plant parts. Notably, the flower had the second highest lignan content, with a value of 5.62 ± 0.33 mg/g·DW. The most commonly used part, the pulp, had a similar lignan content (3.20 ± 0.44 mg/g·DW) to that of the leaves (3.67 ± 0.21 mg/g·DW), and a lower content than the flower. The lignan content was lowest in the stem, with a value of 2.22 ± 0.19 mg/g·DW. This result is consistent with a previous report that the leaf of *S. chinensis* may also be a good source of lignans [11].

Schisandrol A (**1**) was the most abundant of the seven lignans in the flower, pulp, and seed, while the content of schisandrol B (**2**) and schisandrin B (**6**) was highest in the stem and leaf, respectively. Schisandrol A (**1**) and schisandrin B (**6**) were found to be the major lignans present in the pulp and seed of the Omija fruit. The content of these compounds in the seed was ten times higher than that in the other plant parts (Table 2). Tigloylgomisin H (**3**) was not detected in the pulp.

The leaf displayed a higher lignan content than the stem. This result is different from a previous report indicating that the main branch or side branch had a higher lignan content than the leaf [10]. This discrepancy could be ascribed to the difference in the collection season and region, the diameter of the stem, and the habitat [6]. In fact, the stem of the side branch that is usually discarded after pruning was used in this study. It has been reported that the dried seeds have the highest content of schisandrin (schisandrol A, 9.46 mg/g·DW) and total lignan (25.97 mg/g·DW) among the skin, pulp, and seeds of the fruit [12,13]. That report is consistent with the results of this study. However, the total lignan content of the seed in our study was approximately twice as high as that found in previously reported. This discrepancy may be ascribed to the different samples and preparation methods. Soxhlet extraction at 80 °C for 3 h or supercritical CO*_2_* was used in the previous reports [12,13]. In the seeds collected in Europe, the content of gomisin N (**6**) was comparable to or half of that of schisandrol A (**1**) [13].

In previous reports, the lignan content of the fruit showed similar patterns in the samples collected or cultivated in Asia and Europe [7,13,14,15]. The content of schisandrol A (**1**) was the highest, followed by the content of gomisin N (= schisandrin B) (**6**) and schisandrol B (**2**) in order. This tendency in the lignan content of the berry was also shown in direct analysis in real time ionization source coupled with quadrupole orbitrap mass spectrometry as well as in our study [16]. The total lignan contents (%, weight/dry fruit) were in the range of 1.3 to 2.8% in the variable experimental conditions, and the value in this study was 2.7%. From these results, it can be suggested that ultrasonic extraction with methanol at room temperature is the best method for extraction of lignan compounds in the Omija berry.

### 2.2. Lignan Content of Omija Beverage during Fermentation Process

To investigate the changes in the content of seven lignans during the fermentation of the Omija beverage, the most popular food product of this fruit, the raw fruit with skin, pulp, and seed was fermented with white sugar for 12 months. The changes in the lignan composition were monitored at 3, 7, 10, and 12 months during the year-long fermentation process. The total content of the seven lignans after 12 months increased to a maximum value of 2.6 times compared to that after 3 months (Table 3). The lignan content increased rapidly up to 7 months and increased gradually thereafter up to 12 months. Six lignans, excluding angeloylgomisin H (**4**), increased as the fermentation period increased. Angeloylgomisin H (**4**) increased up to 7 months and decreased considerably by 12 months.

Thus far, there is no report on the change in the content of dibenzocyclooctadiene-type lignans in Omija beverages made from the whole fruits of *S. chinensis* with white sugar according to the fermentation period. The fermentation process with sugar of 50 Brix is the most popular method used in Korea. Our study results suggest that a fermentation period of 12 months could be considered as the optimal time for highly efficient extraction or bioconversion to bioactive compounds. Among the seven lignans, schisandrin B (**6**) showed the highest content in the Omija beverage made from fresh berries with white sugar, with a content 7.4- to 12.1-fold higher than that of schisandrol A (**1**) during the fermentation period (Table 3). In the methanol extract of dried fruit of *S. chinensis*, the content of schisandrin A (**1**) was the highest and was 2.0-fold higher than that of schisandrin B (**6**) (Table 2). Schisandrin B (gomisin N) has hepatoprotective, nephroprotective, anti-cancer, anti-aging, and anti-obesity activities [3,4,5]. Schisandrol B (gomisin A) and schisandrin B (gomisin G) have anti-cancer and anti-aging activities [3]. Schisandrin A inhibits dengue viral replication [22]. These lignans are known to be absorbed in rat everted gut sac, human Caco-2 cell monolayer, and in vivo rat models [23]. These reports support the fact that the major lignans in Omija beverages fermented with sugar, a commonly consumed product, would be absorbed by the body and exert various biological activities.

Meanwhile, the study of natural deep eutectic solvents (NADES) has revealed that certain mixtures of natural products, such as sugars, organic acids, and amino acids, in the solid state with the proper ratio, become liquid and enhance the solubility or stability of non-water soluble bioactive compounds [24,25]. Based on this previous report, the high concentration of sugar and organic acids in the berry of *S. chinensis* in this study might play a role in promoting dissolution of the lignan compounds.

### 2.3. Lignan Content in Omija Beverages and Drink from S. chinensis and Sugars

Raw Omija fruit was fermented with three kinds of fermentable sugars (45‒50 Brix) or immersed in alcohol (30%, v/v) for 12 months at room temperature, and the contents of seven lignans in the Omija beverages and drink were evaluated. The three kinds of fermentable sugars were white sugar, brown sugar, and oligosaccharide/white sugar (1:1). The Omija drink prepared by immersing the fruit in alcohol had a higher total lignan content (257.37 μg/g FW) than the three Omija beverages fermented with sugars (Table 4). This result is attributed to the higher solubility of lignan compounds in alcohol than in water. Among the three Omija beverages, the Omija beverage fermented with oligosaccharide/white sugar (1:1, *w*/*w*) had the highest total lignan content, which was 1.2- and 1.7-fold higher than those of the beverages prepared by fermentation with white sugar and blown sugar, respectively. For the Omija drink prepared by immersion in alcohol, the content of compounds 1‒4 was relatively higher than that in the Omija beverages fermented with sugars (3‒5-fold). For all samples, the content of compounds 5‒7 did not change significantly during the total fermentation period. For the Omija beverage fermented with brown sugar, the content of schisandrin B (**6**) and schisandrin C (**7**) was relatively low compared to that in the other beverages.

After the fermentation process, the sugar content was reduced to 13.5, 13.9, and 12.1 Brix for the beverages fermented with white sugar, brown sugar, and oligosaccharide/white sugar, respectively, compared to the initial concentrations. This result is consistent with the previous result in which sucrose was detected before fermentation, but was not detected in the fermented beverage [17]. Among the three kinds of fermentable sugars used in this study, oligosaccharide/white sugar might have acted as the most efficient NADES for dissolution of the lignan compounds in the Omija berry. While there is no study on the bioconversion of dibenzocyclooctadiene-type lignans, a study on the bioconversion of cereal enterolignans by in vitro fermentation for 24 h showed that the total amount of enterolignans formed was closely correlated with the presence of hexoses and pentoses [26]. Studies on the bioconversion of plant lignans to enterolignans in the human gut were reported with various plants lignans, such as secoisolariciresinol and sesaminol [27,28]. Previous studies on the use of fructo-oligosaccharides as prebiotics showed several beneficial effects, including improving mineral absorption, decreasing serum cholesterol and triglycerides, and enhancing the growth of beneficial bacteria in the colon [29]. These reports suggest that intake of Omija beverages fermented with fructo-oligosaccharide can afford further health benefits.

This fermentation process is commonly used in Korea and is not affected by specific microbial organisms, such as yeast, bacteria, or mold, due to the osmotic effect of the high sugar concentration (about 50 Brix), which inhibits growth of the microbes; the process conditions are not controlled and specialized equipment is not required [30]. Recently, the importance of a green chemistry strategy has emerged due to the environmental sustainability and the minimalization of harmful chemical exposure [31]. Although further research is needed, this fermentation method employing a high sugar concentration is recommendable in terms of nontoxicity, eco-friendliness, and ready availability of the required systems.

## 3. Materials and Methods

### 3.1. Plant Materials

The stem, leaf, and flower of *S. chinensis* were collected in May 2016, and whole fruits were collected in August 2016, from Geochang province of South Korea. Each part of the plant was washed with tap water and dried in the shade. This plant was identified by Prof. Mi-Jeong Ahn, College of Pharmacy, Gyeongsang National University, and the voucher specimens were deposited in the herbarium of the College of Pharmacy, Gyeongsang National University (PGSC420–PGSC423). The fruit pulp was peeled off from the whole fruit to prepare the seeds separately. All samples were freeze-dried at −55 °C (FDB-5503, Operon, Kimpo, Korea) and stored at −80 °C before analysis.

### 3.2. Chemicals and Materials

Water and methanol used in the HPLC system were HPLC grade (Fisher Scientific Korea Ltd., Korea), and the other chemicals were of extra grade (Junsei Chemical Co. Ltd., Tokyo, Japan). Seven lignans for reference, including schisandrol A, schisandrol B, tigloylgomisin H, angeloylgomisin H, schisandrin A, schisandrin B, and schisandrin C were isolated and provided by Prof. Chul Young Kim, College of Pharmacy, Hanyang University [32].

### 3.3. Fermentation of the Omija Fruit with Sugars and Alcohol

First, 10 kg of raw fruit was fermented with white sugar (1:1, *w*/*w*) in a porcelain container placed outdoors. An aliquot of the resulting beverage was collected after 3, 7, 10, and 12 months and filtered. Separately, 1 kg of raw fruit was fermented for 12 months in three different ways using 1 kg each of white sugar, brown sugar, and fructo-oligosacchrides/white sugar (1:1, *w*/*w*). A transparent glass jar with a lid was used as the container for fermentation. After keeping the jar at room temperature during the indicated times, an aliquot of the sample was collected and filtered with gauze. The filtrate was kept at −20 °C before the analysis. The fermentation with alcohol was started with 1 L of commercial 30% ethanol for food processing. After 12 months, an aliquot of the sample was collected and filtered with gauze. The filtrate was used for compositional analysis of the lignans. The Brix of the sample was measured three times using a Pocket Refractometer PAL-1 (Atago Co., Ltd., Tokyo, Japan) with a range of 0.0–53.0% Brix. Refractometer calibration and sample readings were performed according to the manufacturer’s instructions.

### 3.4. Sample Preparation

To prepare the sample, 1 g of freeze-dried pulp, seed, leaf, and stem, and 200 mg of flowers were ground and extracted under sonication with 200 mL of methanol for 2 h. The extracts were centrifuged at 3000 g at 4 °C for 10 min, and the supernatant was filtered with a 0.45 μm PTFE membrane filter (Whatman, New York, NY, USA) for HPLC analysis.

Subsequently, 100 μL of filtrate obtained from each Omija beverage was diluted with 200 μL of MeOH. Thereafter, 200 μL of the diluted sample was loaded on Sep-Pak Cartridges (C18, Waters Corporation, Milford, MA, USA) and then eluted with 3 mL of MeOH. The eluent was analyzed by HPLC.

### 3.5. Lignan Analysis

An Agilent 1100 HPLC system (Hewlett-Packard, Waldbronn, Germany) equipped with an autosampler, a column oven, and a diode array detector was used to determine the content of the lignans. Standard solution and sample extracts were injected on a Phenomenex Luna C18 column (4.5 × 150 mm, 5 μm, Torrance, CA, USA) with the volume of 20 μL. The flow rate was 0.8 mL/min, and the column temperature was maintained at 30 °C. The mobile phase was a mixture of water (A) and methanol (B). Solvent (A) was kept at 90% for the first 5 min, then changed from 90% to 10% for next 30 min, and finally maintained at 90% for 5 min. The eluent was detected at wavelength 254 nm. Chemstation software (Agilent Technologies, Santa Clara, CA, USA) was used to operate this HPLC system and to manipulate the data.

Peaks of seven lignans were quantified by an external standard method. Seven lignans were separately dissolved in methanol, and serial dilution was performed to obtain final concentrations of 100, 50, 25, 12.5, 6.25, 3.13, 1.56, 0.78, 0.20, and 0.05 μg/mL. At this chromatographic condition, peaks of the standard lignans were detected at the following tR (min): 11.4 for schisandrol A, 12.9 for schisandrol B, 13.5 for tigloylgomisin H, 14.1 for angeloylgomisin H, 21.1 for schisandrin A, 24.4 for schisandrin B, and 27.0 for schisandrin C (Figure 1 and Figure 2A). The content of lignans in the fruit was calculated from the experimental data of pericarp and seed.

### 3.6. Statistical Analysis

All experiments were performed in triplicate, and the data were expressed as mean ± SD (standard deviation) in Excel software. The one-way ANOVA (analysis of variance) was evaluated by the SPSS statistics 21.0 (IBM Corp, Armonk, NY, USA) using Dunnett’s test. The statistically significance was considered at the value of *p* < 0.05.

## 4. Conclusions

The content of seven lignans in various plant parts of *S. chinensis*, including the stem, flower, leaves, seeds, and pulp, was determined using an HPLC-DAD method. The total lignan content followed the descending order of seed, flower, leaves, pulp, and stem. Analysis of seven major lignans during the fermentation of the Omija fruit with white sugar revealed that the lignan content increased by ~2.4-fold after seven months’ fermentation compared to that at three months’ fermentation, indicating that the lignan content changed in proportion to the fermentation period. The content of total lignans in the Omija beverage fermented with oligosaccharide/white sugar was approximately 1.2- and 1.7-fold higher than that of the Omija beverages fermented with white sugar and brown sugar, respectively. These results provide scientific background illuminating the eco-friendly food processing method using a high sugar concentration that promotes dissolution of the naturally occurring substances as well as the usage of other plant parts of *S. chinensis* as functional materials.

## Figures and Tables

**Figure 1 plants-10-00361-f001:**
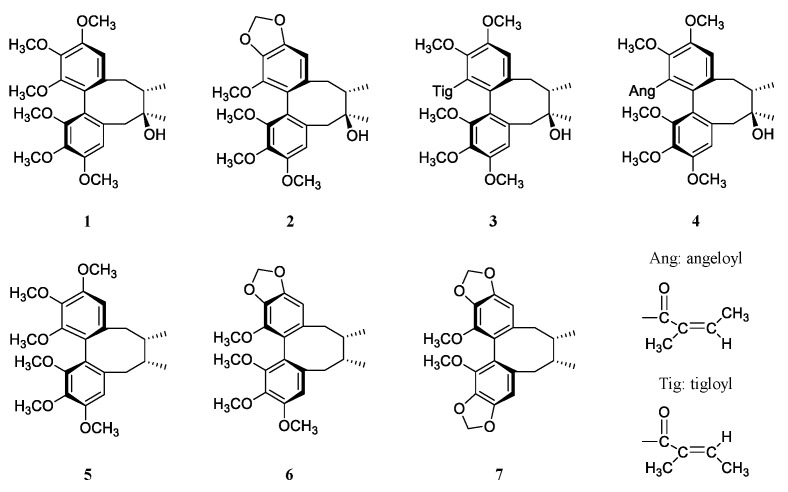
Chemical structures of seven dibenzocyclooctadiene-type lignans (**1**–**7**). **1**, schisandrol A; **2**, schisandrol B; **3**, tigloylgomisin H; **4**, angeloylgomisin H; **5**, schisandrin A; **6**, schisandrin B; **7**, schisandrin C.

**Figure 2 plants-10-00361-f002:**
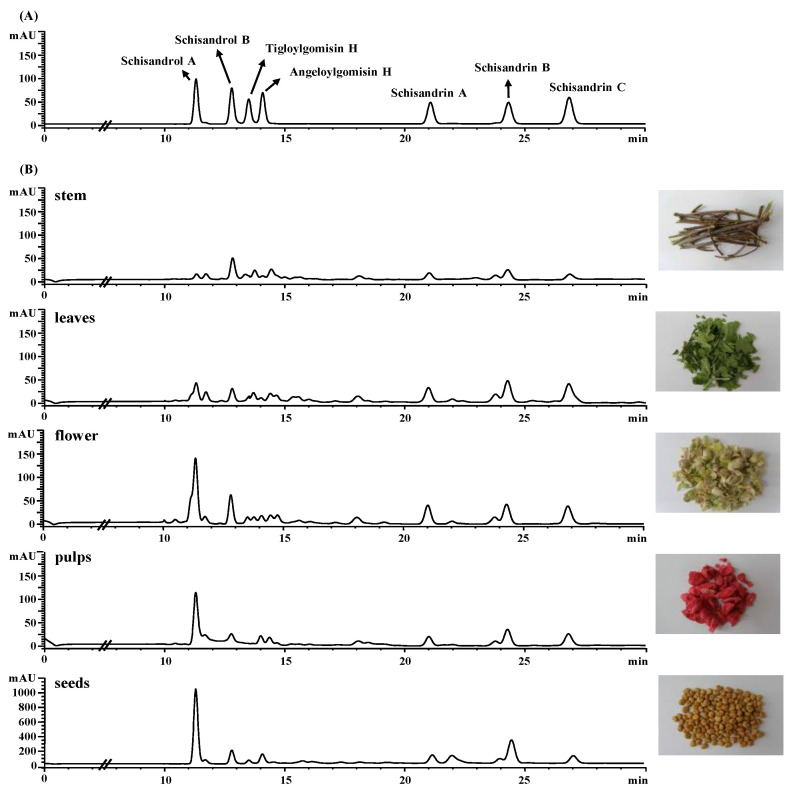
HPLC chromatograms of standard lignans (**A**) and sevens lignans in various parts of *S. chinensis* (**B**) measured at 254 nm.

**Table 1 plants-10-00361-t001:** Linear ranges and correlation coefficients of calibration curves.

Lignans	Range (μg/mL)	Slope (*a*) ^a^	Intercept (*b*) ^b^	Regression (*r^2^*)	Limits of Detection (LOD) (ng)
Schisandrol A (**1**)	0.05–50	226.12	−65.534	0.9974	~0.3
Schisandrol B (**2**)	0.05–12.5	241.27	−23.306	0.9982	~0.3
Tigloylgomisin H (**3**)	0.05–25	174.57	−22.081	0.9995	~0.3
Angeloylgomisin H (**4**)	0.05–25	179.53	−22.650	0.9993	~0.3
Schisandrin A (**5**)	0.05–50	195.70	−74.614	0.9986	~0.3
Schisandrin B (**6**)	0.05–50	251.32	−96.463	0.9985	~0.3
Schisandrin C (**7**)	0.05–12.5	520.59	39.137	0.9986	~0.3

^a,b^ Slope and intercept are represented by a and b, respectively, in the Y= ax + b linear model. Y is peak area and x is concentration.

**Table 2 plants-10-00361-t002:** Contents of seven lignans in plant parts of *S*. *chinensis.*

Parts	1	2	3	4	5	6	7	Total
Stem	0.28 ± 0.05 ^f^	0.57 ± 0.04 ^c^	0.28 ± 0.02 ^c^	0.15 ± 0.02 ^e^	0.36 ± 0.04 ^f^	0.47 ± 0.03 ^f^	0.12 ± 0.00 ^f^	2.22 ± 0.19 ^e^
Leaf	0.73 ± 0.06 ^e^	0.38 ± 0.02 ^d^	0.27 ± 0.01 ^c^	0.18 ± 0.01 ^e^	0.78 ± 0.04 ^d^	0.91 ± 0.05 ^c^	0.43 ± 0.02 ^c^	3.67 ± 0.21 ^b^
Flower	1.85 ± 0.12 ^c^	0.57 ± 0.02 ^c^	0.29 ± 0.01 ^c^	0.36 ± 0.02 ^c^	1.08 ± 0.07 ^c^	1.09 ± 0.06 ^d^	0.39 ± 0.03 ^d^	5.62 ± 0.33 ^c^
Pulp	1.33 ± 0.24 ^d^	0.26 ± 0.05 ^e^	ND	0.28 ± 0.04 ^d^	0.46 ± 0.03 ^e^	0.66 ± 0.04 ^e^	0.23 ± 0.03 ^e^	3.20 ± 0.44 ^d^
Seed	21.0 ± 1.00 ^a^	4.02 ± 0.36 ^a^	1.66 ± 0.28 ^a^	3.96 ± 0.32 ^a^	4.43 ± 0.24 ^a^	10.6 ± 0.52 ^a^	1.74 ± 0.09 ^a^	47.42 ± 2.81 ^a^
Fruit	11.9 ± 0.54 ^b^	2.27 ± 0.23 ^b^	0.89 ± 0.18 ^b^	2.26 ± 0.18 ^b^	2.60 ± 0.18 ^b^	6.02 ± 0.36 ^b^	1.04 ± 0.05 ^b^	26.99 ± 1.7 ^b^

Data are expressed as the mean (the average value of content for dry weight, mg/g DW) and SD of three independent experiments. Different letters in the same column indicate significant differences (*p* < 0.05) between the values. ND means not detected. **1**, schisandrol A; **2**, schisandrol B; **3**, tigloylgomisin H; **4**, angeloylgomisin H; **5**, schisandrin A; **6**, schisandrin B; **7**, schisandrin C.

**Table 3 plants-10-00361-t003:** Contents of seven lignans in an Omija beverage fermented with white sugar during different fermentation times.

Period (Month, m)	1	2	3	4	5	6	7	Total
3 m	7.01 ± 1.30 ^c^	1.65 ± 0.29 ^c^	1.03 ± 0.13 ^c^	2.34 ± 0.44 ^bc^	0.90 ± 0.16 ^b^	51.71 ± 5.14 ^b^	0.65 ± 0.10 ^a^	65.30 ± 7.55 ^b^
7 m	10.42 ± 1.10 ^b^	2.39 ± 0.07 ^b^	1.63 ± 0.14 ^b^	4.33 ± 0.40 ^a^	1.64 ± 0.11 ^a^	126.43 ± 19.53 ^a^	0.69 ± 0.07 ^a^	147.53 ± 21.42 ^a^
10 m	11.96 ± 0.65 ^a^	2.79 ± 0.29 ^a^	1.68 ± 0.16 ^ab^	2.49 ± 0.11 ^b^	1.73 ± 0.20 ^a^	133.58 ± 19.30 ^a^	0.75 ± 0.07 ^a^	154.97 ± 20.78 ^a^
12 m	13.24 ± 2.27 ^a^	2.87 ± 0.52 ^a^	1.95 ± 0.30 ^a^	1.96 ± 0.36 ^c^	2.00 ± 0.47 ^a^	147.23 ± 33.09 ^a^	0.89 ± 0.23 ^a^	170.14 ± 37.23 ^a^

Data are expressed as the mean (the average value of content for fresh weight, μg /g FW) and SD of three independent experiments. Different letters in the same column indicate significant differences (*p* < 0.05) between the values. **1**, schisandrol A; **2**, schisandrol B; **3**, tigloylgomisin H; **4**, angeloylgomisin H; **5**, schisandrin A; **6**, schisandrin B; **7**, schisandrin C.

**Table 4 plants-10-00361-t004:** Contents of seven lignans in Omija beverages or drink fermented with different sugars.

Sugars	1	2	3	4	5	6	7	Total
White sugar	10.34 ± 1.44 ^c^	2.80 ± 0.21 ^b^	1.78 ± 0.35 ^c^	4.69 ± 0.23 ^b^	1.93 ± 0.05 ^c^	151.78 ± 14.89 ^b^	0.89 ± 0.03 ^b^	174.50 ± 16.00 ^c^
Brown sugar	6.24 ± 0.11 ^d^	1.74 ± 0.20 ^c^	0.94 ± 0.01 ^c^	3.52 ± 0.12 ^c^	2.60 ± 0.03 ^b^	112.09 ± 5.15 ^c^	0.63 ± 0.07 ^c^	127.76 ± 3.81 ^d^
Oligosaccharide/white sugar (1:1)	16.58 ± 1.07 ^b^	3.38 ± 0.60 ^b^	4.03 ± 0.76 ^b^	3.85 ± 0.49 ^c^	2.49 ± 0.42 ^b^	183.01 ± 23.89 ^a^	1.01 ± 0.13 ^a^	214.14 ± 18.69 ^b^
Alcohol	36.66 ± 1.57 ^a^	6.13 ± 0.67 ^a^	5.16 ± 0.90 ^a^	6.69 ± 0.42 ^a^	2.76 ± 0.12 ^a^	199.02 ± 16.23 ^a^	0.95 ± 0.02 ^a^	257.37 ± 13.31 ^a^

Data are expressed as the mean (the average value of content for fresh weight, μg /g FW) and SD of three independent experiments. Different letters in the same column indicate significant differences (*p* < 0.05) between the values. **1**, schisandrol A; **2**, schisandrol B; **3**, tigloylgomisin H; **4**, angeloylgomisin H; **5**, schisandrin A; **6**, schisandrin B; **7**, schisandrin C.

## Data Availability

Not applicable.

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
