# Peer review of "Dibenzocyclooctadiene Lignans in Plant Parts and Fermented Beverages of Schisandra chinensis"

_plants, 2021, doi:10.3390/plants10020361_

Round 1

Reviewer 1 Report

The content of seven lignans in plant parts of S. chinensis like the stem, flower, leaves, seeds and pulp was determined using a method based in HPLC-DAD.

Content of lignan was assessed and compared; seeds contained the highest amount.

Fermentation conditions were applied of the Omija fruit, and lignans content were determined.

In brief, it seems that the strategy according to the objectives of this study is appropriate. However, I believe that the time for fermentation is too long; and that researches should be done to shorten the so long fermentation periods. Some comments on this matter are required; they should be mentioned in Abstract and in Conclusions.

There is no any very recent reference cited in this work. Authors should try to cite recent studies, or just one, published in 2020.

Final suggestion: to accept this study for publication with MINOR changes.

Author Response

Thank you very much for your consideration. We have considered your comments (Original Reviewer’s Comments, ORC) very closely in revising this manuscript as follows.

ORC:

In brief, it seems that the strategy according to the objectives of this study is appropriate. However, I believe that the time for fermentation is too long; and that researches should be done to shorten the so long fermentation periods. Some comments on this matter are required; they should be mentioned in Abstract and in Conclusions.

-->The description of “In general, people make this beverage at home and consume the beverage after 6 or 12 months or much longer period fermentation.” was added in the Introduction Section to clarify the reason that 12 month-long fermentation period had been selected in this study (in Page 2, lines 61-63).

 ORC:

There is no any very recent reference cited in this work. Authors should try to cite recent studies, or just one, published in 2020.

-->Two very recent references published in 2020 and 2021, respectively, were cited (in Page 4, lines 142 and 147; Reference #, 15 and 16).

Thank you again.

Reviewer 2 Report

Manuscript entitled „Dibenzocyclooctadiene Lignans in Plant Parts and Fermented Beverages of Schisandra chinensis” deals with the estimation of the content of only 7 (main) dibenzocyclooctadiene lignans in the stem, leaf, pulp, seed, and flower of S. chinensis. This aspect of work is not new; these compounds were estimated in the Sch plant material by many authors before. I recommend to add the comparison of the own studies with former ones eg. as Table where your results could be described as Sch of Koran origin and could be compared to the materials of other origin eg. Chinese, Japanese, European. The interesting aspect of work is concerned on the estimation of lignans in Omija beverage made from the whole fruit fermented with white sugar for different fermentation times. But these studies also are not innovative. There are other similar studies performed before. Please discuss more your results. Authors proved the change in the lignan content is compared for the fruit beverage fermented with three kinds of sugars (white, brown, and oligosaccharide/white) and ethanol and that is the most interesting. The content of total lignans in Omija beverage fermented with white sugar was approximately 1.2- and 1.7-fold higher than that of Omija beverage fermented with white sugar and brown sugar, respectively. These results were indicated as new method of  eco-friendly food processing using a high sugar concentration that promotes dissolution of the lignans. Can you give more advantages for your results? One question more - Why you studied lignans content in different parts of plants while the work is concernd on fruits – beverages from fruits? Why beverages from other plant parts weren’t studied? That could be interesting…

Author Response

Thank you very much for your careful and thorough reading of this manuscript and for the thoughtful comments and constructive suggestions, which helps to improve the quality of this manuscript. We have considered your comments (Original Reviewer’s Comments, ORC) very closely in revising this manuscript as follows.

ORC:

Schisandra chinensis” deals with the estimation of the content of only 7 (main) dibenzocyclooctadiene lignans in the stem, leaf, pulp, seed, and flower of S. chinensis. This aspect of work is not new; these compounds were estimated in the Sch plant material by many authors before. I recommend to add the comparison of the own studies with former ones eg. as Table where your results could be described as Sch of Koran origin and could be compared to the materials of other origin eg. Chinese, Japanese, European. The interesting aspect of work is concerned on the estimation of lignans in Omija beverage made from the whole fruit fermented with white sugar for different fermentation times. But these studies also are not innovative. There are other similar studies performed before. Please discuss more your results. Authors proved the change in the lignan content is compared for the fruit beverage fermented with three kinds of sugars (white, brown, and oligosaccharide/white) and ethanol and that is the most interesting. The content of total lignans in Omija beverage fermented with white sugar was approximately 1.2- and 1.7-fold higher than that of Omija beverage fermented with white sugar and brown sugar, respectively. These results were indicated as new method of eco-friendly food processing using a high sugar concentration that promotes dissolution of the lignans. Can you give more advantages for your results? One question more - Why you studied lignans content in different parts of plants while the work is concerned on fruits – beverages from fruits? Why beverages from other plant parts weren’t studied? That could be interesting…

  • The comparison of our own studies with previously reported former ones was added in the subsection 2.1 of the Results and Discussion Section (in Pages 4 and 5, lines 139-150). It included comparison on the origin and experimental conditions.
  • To the best of our knowledge, there is no similar previous report on Omija whole fruit fermented with white sugar or other reports which are worth to be mentioned.
  • More advantage of our study result were added with the advantage of oligosaccharide/white sugar in the subsection 2.2 of the Results and Discussion section (in Pages 5-6, lines 180-191 and in Page 6-7, lines 226-228).
  • The main reason for this study on lignan contents in different parts of S. chinensis is included in the Introduction Section (in Page 2, lines 53-57). Based on our results, the scientific background for the use of other parts except the berry could be suggested. The raw berry only is mainly converted to a fermented beverage with sugar as the most common form for consumption in Korea (in Page 2, lines 60-61). This is the reason that the berry beverages only were evaluated in this study. The study on the beverage made from other parts is worth to be accomplished in the future.

Thank you again.

Reviewer 3 Report

The Authors proposed an analytical method to determine the composition in 7 dibenzocyclooctadiene lignans in 5 different parts of S. chinensis plants as well as in fermented beverages obtained after different fermentation times or the used of different sugars for fermentation process using an ecofriendly method.

This paper is interesting, and I judge that the presented results deserved publication in Plants.

I have some minor revisions to suggest:

- Indicate the limit of quantification (LOQ)

- S. chinensis in italics (eg Table 2 legend or lines 151 and 160)

- line 161 lignan compounds instead of lignin compounds

- improve discussion about dibenzocyclooctadiene lignans production during fermentation procedure in particular for the different sugar used. For example, Nowak et al indicate antimicrobial activity for some dibenzocyclooctadiene lignans from S. chinensis (Nowak, A., Zakłos-Szyda, M., Błasiak, J., Nowak, A., Zhang, Z., & Zhang, B. (2019). Potential of Schisandra chinensis (Turcz.) Baill. in human health and nutrition: a review of current knowledge and therapeutic perspectives. Nutrients, 11(2), 333.), whereas Yang et al have proposed some metabolization of dibenzocyclooctadiene lignans from S. chinensis (Yang, J. M., Siu-po, P., Hok-keung, J. Y., & Che, C. T. (2011). HPLC-MS analysis of Schisandra lignans and their metabolites in Caco-2 cell monolayer and rat everted gut sac models and in rat plasma. Acta Pharmaceutica Sinica B, 1(1), 46-55.) that could be taken into account.

- Put Figure 2 and its legend on the same page

- 3.5. HPLC procedure: indicate the injection volume

Author Response

Thank you very much for your consideration. We have considered your comments (Original Reviewer’s Comments, ORC) very closely in revising this manuscript as follows.

ORC:

- Indicate the limit of quantification (LOQ)

- S. chinensis in italics (eg Table 2 legend or lines 151 and 160)

- line 161 lignan compounds instead of lignin compounds

  • The limits of quantification (LOQ) were indicated in the Results and Discussion Section (in Page 2, line 86).
  • All descriptions of “S. chinensis” were changed into italic.
  • Lignin typos were corrected to lignan.

ORC:

- improve discussion about dibenzocyclooctadiene lignans production during fermentation procedure in particular for the different sugar used. For example, Nowak et al indicate antimicrobial activity for some dibenzocyclooctadiene lignans from S. chinensis (Nowak, A., Zakłos-Szyda, M., Błasiak, J., Nowak, A., Zhang, Z., & Zhang, B. (2019). Potential of Schisandra chinensis (Turcz.) Baill. in human health and nutrition: a review of current knowledge and therapeutic perspectives. Nutrients, 11(2), 333.), whereas Yang et al have proposed some metabolization of dibenzocyclooctadiene lignans from S. chinensis (Yang, J. M., Siu-po, P., Hok-keung, J. Y., & Che, C. T. (2011). HPLC-MS analysis of Schisandra lignans and their metabolites in Caco-2 cell monolayer and rat everted gut sac models and in rat plasma. Acta Pharmaceutica Sinica B, 1(1), 46-55.) that could be taken into account.

  • More discussion about dibenzocyclooctadiene lignans production during fermentation procedure were added in 2.2 subsection of the Results and Discussion section (in Pages 5-6, lines 180-191). The related description includes the reviewer’s suggested references.

ORC:

- Put Figure 2 and its legend on the same page

- 3.5. HPLC procedure: indicate the injection volume

  • The positions of Figure 2 and its legend were moved to be put on the same page (in Page 3).
  • The injection volume was added in 3.5 subsection of the Materials and Methods Section (in Page 9, lines 297-298). 

Thank you again.

Round 2

Reviewer 2 Report

All my recomendations were applied.